# Preparation of Pickering Emulsions Stabilized by Modified Silica Nanoparticles via the Taguchi Approach

**DOI:** 10.3390/pharmaceutics14081561

**Published:** 2022-07-28

**Authors:** Fatemeh Heidari, Seid Mahdi Jafari, Aman Mohammad Ziaiifar, Nicolas Anton

**Affiliations:** 1Faculty of Food Science and Technology, Gorgan University of Agricultural Sciences and Natural Resources, Gorgan 4913815739, Iran; fatemeh_heidari87@yahoo.com (F.H.); amziaiifar@gau.ac.ir (A.M.Z.); 2INSERM (French National Institute of Health and Medical Research), UMR 1260, Regenerative Nanomedicine (RNM), FMTS, University of Strasbourg, F-67000 Strasbourg, France; nanton@unistra.fr

**Keywords:** silica nanoparticles, surface modification, Pickering emulsion, optimization, physical stability

## Abstract

In this study, oil-in-water Pickering emulsions (PEs) were prepared by modified silica nanoparticles (MSNs) with cetyltrimethylammonium bromide (CTAB) using the Taguchi approach. The surface modification of SiO_2_ nanoparticles (NPs) was performed in different conditions, temperatures, pH levels, and amounts of CTAB as a coating agent, followed by an evaluation of their physicochemical properties. After treatment of the SiO_2_ NPs, the relationship of the MSNs’ surface properties and their efficiency in stabilizing Pickering emulsions was investigated by considering the zeta potential (ZP) and emulsion physical stability as main responses, respectively. Results disclosed were then supported by additional characterization, such as thermogravimetric analysis (TGA), Fourier transform infrared (FTIR) spectroscopy, contact angle (CA), and scanning electron microscopy. Results demonstrated that temperature has the most important role in the treatment of SiO_2_ nanoparticles, and allows for the identification of the best experimental conditions, i.e., range of zeta potential of MSNs to produce more efficient NPs, as well as the best stabilization of PEs.

## 1. Introduction

In Pickering emulsions (PEs), surfactants are replaced by colloidal particles to stabilize the interface between the oil and water phases [1,2]. These systems display greater physical/chemical stabilities, such as long-term stability and high resistance to coalescence compared with surfactant-stabilized emulsions [3]. This arises from the fact that more energy is required to remove the particles from the liquid–liquid interface relative to desorbing the surfactants [4]. These systems are therefore widely used in various food, pharmaceutical, and cosmetic industries [5]. Different inorganic and organic particles can be employed as stabilizers for PEs, including SiO_2_, CaCO_3_, BaSO_4_, clays, carbon black, carbon nanotubes, cellulose, starch, and chitosan [6,7].

SiO_2_ nanoparticles (NPs), also called nano-silica, are extensively applied due to their cost-effectiveness, biocompatibility, low in vivo toxicity, and antimicrobial properties. However, they cannot be directly used to stabilize PEs due to the presence of plentiful hydroxyl groups (–OH) on their surface, which cause the NPs to have a strong degree of hydrophilicity and low surface activity [8,9,10]. Accordingly, the surface of SiO_2_ NPs should be modified to achieve a proper degree of hydrophobicity and optimum wettability at the oil–water (O/W) interface [2,5]. Properties of SiO_2_ NPs can be altered by chemical and/or physical methods [11]. Physical modification is more noticeable since it is more straightforward and has a lower experimental cost in comparison to chemical modification [12]. Physical modification is induced from physical interactions between surfactants (or large molecules) and NPs, resulting in the modification of the surface properties of NPs, which can lead to greater adsorption of NPs at the O/W interface, as well as high stability [9,13,14].

Numerous studies have previously focused on particle surface modification through the adsorption of surface-active agents (i.e., surfactants, proteins, and polymers), e.g., chitosan, alginate, and cetyltrimethylammonium bromide (CTAB) for polystyrene, silica, and bentonite, respectively [15,16,17]. CTAB is a cationic surfactant widely employed in the pharma industry. When CTAB is mixed with anionic stabilizers such as silica and bentonite nanoparticles, electrostatic interaction between their surface charges increases the adsorption of CTAB molecules on the surface of nanoparticles. As a result, this can lead to more surface activity and strongly affect their hydrophilic/hydrophobic properties [18]. Some systematic studies on the surface modification of SiO_2_ NPs with CTAB at water–air [19], water–hexane [20], water–n heptane [21], and wax–water [22] interfaces are available. These authors obtained an increase in the NPs’ hydrophobicity, attributed to the electrostatic interactions between the negatively charged surface of SiO_2_ NPs and the positively charged surface of CTAB [19,21]. As a result, NPs are coated with hydrophilic moieties of CTAB. On the other hand, researchers have also demonstrated that surface modification of NPs directly depends on several factors, such as the properties of NPs and surfactants, the ratio of NPs/surfactant, temperature, and pH [22,23].

The physicochemical parameters of NPs such as size, shape, wettability, and surface charge play a key role in the stability of PEs [24]. The particle surface wettability is the most important factor in determining the type of emulsions related to the hydrophilicity or hydrophobicity of the particles, which is evaluated by measuring the contact angle [25]. Particle size affects the particle mobility in the continuous phase and the droplet size of the emulsion [26]. There are colloidal particles with different shapes such as spherical, ellipsoidal, and rod-like particles; among these, spherical particles are the most widely used particles [27]. The desorption energy of particles with various shapes depends on different parameters. For example, in spherical particles, it depends on the interfacial tension, radius of NPs and contact angle, while in rod particles, it depends on the length, diameter, rigidity, and amphiphilicity [28,29].

Emulsions prepared by NPs with a high or low surface charge are not stable. Therefore, this study aims to first modify the surface of SiO_2_ NPs, and then investigate their application as a stabilizer in the formulation of PEs as a potential delivery system for bioactive compounds. To this end, the optimization of the surface properties of SiO_2_ NPs is performed according to the Taguchi method, investigating the effects of SiO_2_ NPs/CTAB ratios, temperature, and pH. As a result, the main outputs were evaluations of the surface charge, chemical structure, wettability, and performance of MSNs as PE stabilizers. One unique aspect of this study lies in how the experimental approach was carried out, using and optimizing SiO_2_ nanoparticles as an efficient solution to stabilize Pickering emulsions. The multistep process, from the surface modification of SiO_2_ NPs, the formulation of Pickering NEs, and up to the optimization of their stability according to the Taguchi method, also constitutes new insight into the field of pharmaceutical formulation.

## 2. Materials and Methods

### 2.1. Materials

Rice husks and glycerol used for the synthesis of SiO_2_ NPs were purchased from a local rice-processing complex in Gorgan (Golestan, Iran) and the Kian Kaveh Amaze Company (Tehran, Iran), respectively. CTAB, acetic acid, and sodium hydroxide were obtained from Merck, Germany. In addition, the applied sunflower oil for the preparation of PEs was provided from a local market in Gorgan (Golestan, Iran). All other chemicals used in this study were of analytical grade. Distilled and deionized water were employed in experiments.

### 2.2. Preparation of SiO_2_ NPs

SiO_2_ NPs were synthesized from rice husks according to the modified method of Suttiruengwong et al. [30]. Rice husk ash (RHA) was first obtained through calcining the rice husks at 500 °C for 24 h in a furnace (LE 14/11/B150, Nabertherm, Lilienthal, Germany). RHA was then mixed with glycerol as a depolymerizing agent at a ratio of 1:10 *w*/*v* at 200 °C using a magnetic hotplate stirrer for 2 h. The obtained mixture was stored at an ambient temperature for 24 h. Excess glycerol was then evacuated, deionized water was added, and the mixture was stored for 24 h. The obtained gel was washed with distilled water three times and dried at 105 °C for 24 h in an oven (800, Memert, Schwabach, Germany). The NPs were finally formed by calcinating the dried gel at 500 °C for 24 h.

### 2.3. Surface Modification of SiO_2_ NPs

The surface modification of SiO_2_ NPs was performed according to a protocol previously reported [1,20]: 1 g of SiO_2_ NPs was dispersed in 100 mL of deionized water, and the pH of the solution was adjusted at different values by adding acetic acid (1% *v*/*v*) and NaOH (0.5 M). The different treatments are reported in Table 1, according to the Taguchi method; different values of CTAB (powder) were slowly added to suspensions of SiO_2_ NPs at different weight ratios and for different pHs and temperatures under constant stirring. These mixtures were then stirred with a magnetic stirrer for a minimum of 6 h. The suspensions were then centrifuged at 4500 rpm for 10 min, washed with ultrapure water several times, and dried at 70 °C for 24 h in an oven.

### 2.4. Preparation of PEs

To prepare the oil-in-water PEs, 2 mL of sunflower oil was slowly added to 8 mL of MSN suspensions (1%) and then homogenized by a rotor stator homogenizer (SilentCrusher M, Heidolph, Schwabach, Germany) at 14,000 rpm for 6 min.

### 2.5. Determination of Particle Size and Zeta Potential (ZP)

The average diameter, polydispersity index (PDI), and ZP values of SiO_2_ NPs and MSNs were measured using the Zetasizer instrument (Malvern, UK) at 25 °C [22].

### 2.6. Fourier Transform Infrared (FTIR) Spectroscopy

To characterize the chemical interactions, FTIR spectra of SiO_2_ NPs, CTAB, and MSNs were obtained by an FTIR spectrometer (Tensor II, Bruker Spectrometer, Billerica, MA, USA). The samples were mixed with KBr in a ratio of 1:150, and then pressed under a pressure of 4 t, and FTIR spectra were recorded on pressed pellets in the wavenumber range of 500–4000 cm^−1^ [23]. The interactions were analyzed using the Omnic 3.2 software.

### 2.7. Droplet Size Measurement of PEs

The size of droplets in PEs was measured using a dynamic light scattering (DLS) method (Zetasizer, Malvern Instrument, Malvern, UK). The refractive indices of the sunflower oil and deionized water were set at 1.472 and 1.33, respectively [29]. The droplet size of some samples was measured through microscope pictures taken from an optical microscope (Zeisss, Oberkochen, Germany) and analyzed by the Image J software.

### 2.8. Physical Stability of PEs

The storage stability of PEs against separation and coalescence was determined by measuring the emulsion creaming index (*CI*) (Equation (1)) and droplet size, respectively. The samples were sealed into plastic tubes and stored at 25 °C for 30 days.
(1)$$CI\left(\%\right)=\frac{H_{S}}{H_{t}}\;$$

Here, *H_S_* and *H_t_* are the height of the serum layer (mm) and the total emulsion (mm), respectively, as measured with a digital caliper [31].

### 2.9. Thermogravimetric Analysis (TGA)

The weight loss in SiO_2_ NPs and MSNs was determined using a thermogravimetric analyzer (Q600, TA Instruments, New Castle, DE, USA) by heating the samples from 25 to 800 °C at a heating rate of 10 °C/min in the argon atmosphere [20]. Before the test, NPs were dried in the oven at 100 °C for 24 h to remove the excess water on their surface.

### 2.10. Wettability Measurement

An optical contact angle measuring device (OCA15 Plus, Dataphysics, Filderstadt, Germany) was used to characterize the hydrophilic/hydrophobic properties of SiO_2_ NPs and MSNs [22].

### 2.11. Morphological Analysis

Field emission scanning electron microscopy (FESEM) (MIRA III, TESCAN, Ostrau, Czech Republic) was used to determine the microstructure of SiO_2_ NPs; scanning electron microscopy (SEM) (SU3500, Hitachi, Tokyo, Japan) was applied to determine the arrangement of MSNs at the O/W interface (the samples were diluted 20 times, then 1 mL of the obtained solution was dried on aluminum foil) [22].

### 2.12. Statistical Analysis

Statistical analyses were performed by analysis of variance (ANOVA) and significant difference analysis (*p* < 0.05) using the Qualitek software (version 4, Nutek, Albany, NY, USA).

## 3. Results and Discussion

### 3.1. Characteristics of SiO_2_ NPs

The main results on native SiO_2_ NPs are reported in Figure 1. It was revealed that SiO_2_ NPs prepared from rice husks were predominantly spherical, with an average size of 42 ± 0.7 nm and PDI = 0.627 (Figure 1b). ZP measurements of SiO_2_ NPs (Figure 1c) revealed a significantly negative surface charge over a wide pH range (−56.6, −62.2 and −75.7 mV for pH= 5.5, 7.5, and 9.5, respectively) due to abundant hydroxyl groups on their surface. Finally, the wettability of NPs (obtained through the determination of a three-phase CA of water onto a surface coated with NPs; see details in Section 3.6 below) disclosed the total wettability (*θ* = 0°) of the NPs. This confirmed that the surface of the NPs was completely hydrophilic. The set of results emphasized in Figure 1 appears in line with those reported in the literature [22,32,33]. We can conclude that in native form, these SiO_2_ NPs present an important hydrophilicity and affinity to the water phase, mainly due to the high negative charge of their surface, resulting in total wettability with water. The solution that appears to adapt their use as a potential stabilizer for PEs, as presented in the Introduction, plays on the surface charge and composition, an objective that can be achieved through interaction with CTAB.

### 3.2. Zeta Potential of MSNs

ZP appears to be the main parameter determining the surface properties of NPs. Numerous factors can influence this parameter, investigated in preliminary experiments to select the most important factors, and disclosing their working ranges. The experimental ranges of pH, CTAB/SiO_2_ NPs ratios, and temperature were found to be 5.5 to 9.5, 1:8 to 3:8 *w*/*w*, and 30 to 70 °C, respectively (Table 1). The results presented in Figure 2a show the raw ZP values of MSNs, which can vary from −52 to −17 mV. This indicates the effectiveness of the CTAB coating method, and the need to rationalize the selection of process conditions. Treatment No. 7, with CTAB/SiO_2_ NPs ratios of 1:8, pH = 9.5, and T = 70 °C, had the highest ZP (−52 mV), whereas treatment No. 9, with CTAB/SiO_2_ NPs ratios of 3:8, pH = 9.5, and T = 50 °C, led to the lowest ZP (−17 mV).

Based on the obtained data, all independent parameters were effective on the final ZP of SiO_2_ NPs, reported in Figure 2b, illustrating the individual effects of independent factors on the values of ZP. ANOVA results (Figure 2c) showed that pH and temperature were the least (0.782%) and most effective factors (68.688%) on ZP, respectively. As shown in Figure 2a, increasing the modification temperature from 25 to 50 °C decreases the ZP of SiO_2_ NPs, probably due to higher electrostatic interactions between the NPs with surfactant (at the higher temperature) and the increase in the amount of bonded CTAB on the SiO_2_ NPs’ surface. However, a further increase in temperature to 70 °C during the surface modification process resulted in the highest ZP. An explanation for this result is that by exceeding the temperature > 50 °C, a higher amount of CTAB molecules is converted to micelles in water due to the dependence of critical micelle concentration on temperature. The impact of the amount of CTAB on ZP appeared to be rather logical, decreasing the ZP with the increase in surfactants that interact with hydroxyl groups present on the SiO_2_ NPs’ surface. This means that more CTAB has been absorbed on the surface of SiO_2_ NPs.

Our findings were consistent with those of Ma et al. [23] on the surface modification of SiO_2_ NPs by a cationic surfactant, as well as those of Kang et al. [34], reporting similar behavior for the surface modification of Sb_2_O_3_ NPs with CTAB, where the ZP of Sb_2_O_3_ NPs increased significantly at higher CTAB levels. Similarly, although weak, pH induces a decrease in ZP values due to the ionization of –OH groups. However, compared to the effect of temperature, the effect of pH is rather negligible. Thus, the electrostatic interaction between surfactants and SiO_2_ NPs becomes stronger, resulting in lower ZP values of SiO_2_ NPs [1]. These data are completed with the comparison of the variance between the different parameters (Figure 2c) and severity index (Figure 2d). The highest interaction between two independent variables was for the CTAB/SiO_2_ NPs ratio and pH with a severity index of about 38.6%, and the lowest for the CTAB/SiO_2_ NPs ratio and a temperature with an index < 7.7%.

### 3.3. Contact Angle of NPS

In line with the former section on the characterization and optimization of the surface modification of SiO_2_ NPs, another important characterization is the impact of treatments on the wettability of water on a surface composed of such NPs as shown in Figure 1d for native NPs. This was evaluated through the water CA formed over NPs pressed into a tablet form, reported in Figure 3a,b after treatment No. 7 and No. 2, respectively. Many studies have investigated the behavior of various particles at the O/W interface in emulsions by measuring the CA [35,36,37]. Generally, the wetting tendency increases by decreasing the CA < 90°, while it decreases by increasing the CA > 90° [9].

As shown in Figure 3, compared to Figure 1d, the CA changed with the surface modification of SiO_2_ NPs from 0° for native SiO_2_ NPs (indicating that the produced NPs exhibited an absolute hydrophilic behavior) to CA = 85° after treatment No. 2. This confirms the modification of the nature of the SiO_2_ NPs’ surface by CTAB. In addition, *θ* was 13° after treatment No. 7, demonstrating that SiO_2_ NPs were slightly modified and almost completely hydrophilic. Wei et al. [1] examined the effect of different surfactants on SiO_2_ NPs and found a significant effect of surfactant concentrations on surface wettability.

### 3.4. Physical Stability of Pickering Emulsions

After the formulation of PEs, stability study is crucial and confirms the performance of MSNs as a stabilizer [24]. This parameter can be evaluated based on the rate of droplet coalescence, creaming, or emulsion flocculation [29,38]. Considering that the creaming is mostly a precursor to coalescence, the emulsion stability was primarily evaluated visually by measuring the creaming index (CI) as a function of the storage time to acquire the most stable PEs against creaming. Results are reported in Figure 4a, showing the impact of the ZP of MSNs on the CI.

These results showed that freshly prepared PEs were stable and homogeneous, but after 3 days of storage, the lipid-droplet-enriched cream layer was gradually observed for more than half of PEs, and the CI for emulsions stabilized by MSNs with a ZP > −34 mV exceeded 9%. This disclosed a relationship between the surface activity of MSNs and emulsion stability, and subsequently the droplet flocculation and their coalescence. Furthermore, PEs stabilized by MSNs with a ZP = −52 mV had clear phase separation after one day of storage. Their CI quickly increased to 50% after 5 days of storage. In addition, on the 8th day, PEs failed to form, and their structure completely disappeared. This behavior can be attributed to the rapid reduction of adsorbed NPs on the O/W interface during storage and the subsequent formation of dispersed phase droplets with larger sizes and instability of PEs. In contrast, in the sample stabilized by MSNs with a ZP = −26 mV, no phase separation was observed even after one month of storage, and the measured CI values remained close to zero. This result suggests that when MSNs have a high stabilizer efficiency, oil droplet aggregation and creaming is much more inhibited.

As illustrated in Figure 4a, due to the intense coalescence of oil droplets in PEs stabilized by MSNs with a ZP < −34 mV, free oil was observed after 30 days of storage. In contrast, the coalescence in PEs stabilized by MSNs with a ZP > −34 mV appeared negligible and seemed to increase again for the higher ZP (MSN with a ZP = −17 mV). These results brought a range of optimized MSNs for stabilizing PEs. The stability comparison between fresh PEs and those PEs stored for 30 days, reported in Figure 4b, confirmed this optimized formulation range. PEs stabilized by MSN with a ZP = −49 mV displayed significant changes in mean droplet size after 30 days of storage compared to other PEs. Our results indicated the highly charged MSNs contributed to a great degree of instability in PEs, in line with previous studies, e.g., those of de Folter et al. [39] and Hu et al. [29], which similarly reported low stability of PEs prepared by zein and gliadin NPs with an approximate ZP of 60 mV and 25 mV, respectively. NPs with high ZP values produce a repulsive energy barrier when they come close to an O/W interface [40,41]. This can be attributed to the repulsive forces created between NPs, which overcome the convective forces (responsible for transporting them to the O/W interface during emulsification); their adsorption onto the interface is consequently limited, such that unstable emulsions are formed [29].

The CI and coalescence in treatment No. 9 (with the highest ZP) were higher than other PEs stabilized by NPs with a ZP > −34 mV. This phenomenon can be attributed to SiO_2_ NPs modified with more hydrophobicity than other MSNs [42]. In this case, a large surface area of the SiO_2_ NP is modified due to the greater interaction between the hydroxyl groups and the CTAB, leading to the modification of hydrophobicity and hydrophilicity, impacting their PEs’ stabilizer properties. In conclusion, analysis of the results revealed that the optimum conditions to prepare an MSN with suitable wettability, which leads to the formation of a PE with the highest stability, are pH = 5.5, a CTAB/SiO_2_ NPs ratio of 2:8, and a temperature of 50 °C (treatment No. 2).

A complementary characterization was performed (Figure 5), for which SEM images of PEs clearly show the presence of SiO_2_ NPs as a PE stabilizer in the O/W interface. It seems that the NPs’ coverage was very homogeneous and very stable against vacuum conditions undergone during SEM experiments.

### 3.5. Chemical Interactions in NPs

FTIR spectra were studied to evaluate the interactions between SiO_2_ NPs and CTAB (Figure 6), which can be confirmed by the shifts in characteristic peaks and the appearance of new peaks in the MSNs. According to Figure 6, the characteristic absorption peaks at 2849 and 2915 cm^−1^ indicate the typical stretching vibrations of the C–H resulting from the –CH2 and –CH3 groups in the pure CTAB spectrum, respectively [43]. The characteristic peaks in the SiO_2_ NP at 1094, 803, and 3448 cm^−1^ are associated with the stretching vibrations of Si–O, Si–O–Si, and –OH groups, respectively. The adsorption bands for SiO_2_ NPs are consistent with those reported in previous studies [30,43]. The decreased strength of peak 3448 cm^−1^ and the formation of new peaks at 2923 and 2854 cm^−1^ on the surface of MSNs confirm the interaction between SiO_2_ NPs and CTAB.

### 3.6. Thermal Properties of NPs

TGA thermograms of SiO_2_ NPs and MSNs are reported in Figure 7. Although the NPs were dried in the oven, further significant weight loss was observed in the samples. The weight loss at temperatures < 200 °C demonstrated the evaporation of the absorbed water on the surface of NPs. These results showed that SiO_2_ NPs and MSNs had about 5 and 12% weight loss when the temperature was raised > 200 °C. This weight loss can be explained by the dehydration condensation of Si–OH in both samples and the thermal decomposition of CTAB in the MSNs [23]. According to TGA plots, MSNs showed less weight loss compared to SiO_2_ NPs up to 238 °C. This phenomenon could be associated with less water absorption on the surface of MSNs. Accordingly, the TGA results are in good agreement with those obtained from the FTIR.

## 4. Conclusions

In this study, the stabilization of PEs by SiO_2_ NPs was investigated as a function of the surface modification of silica NPs. This modification was undertaken under the Taguchi approach, for which surface modification of SiO_2_ NPs was performed using a coating with CTAB surfactant. Then, such modified MSNs were used to prepare PEs. The independent variables selected in the Taguchi approach were temperature, amount of CTAB respective to that of SiO_2_ NPs, and pH. The ANOVA results revealed that temperature and pH were the most and least effective parameters on the ZP of SiO_2_ NPs, respectively, and that a strong interaction between the CTAB/SiO_2_ NPs ratio and pH were found. According to the findings, increasing the modification temperature from 30 to 50 °C decreased the ZP of SiO_2_ NPs, which could be due to higher electrostatic interactions and the increase in bonded CTAB on the surface of SiO_2_ NPs. The results of FTIR, TGA, and CA confirmed the surface modifications of SiO_2_ NPs. Furthermore, the study of PEs stabilized by MSNs with different ZP values revealed that using MSNs with a ZP = −26 mV was the optimal condition for the best physical stability, obtained with the following treatment: pH = 5.5, the CTAB/SiO_2_ NPs ratio = 2:8, and T = 50 °C. In general, the optimal surface modification conditions of SiO_2_ NPs reported in this work might be expanded to be used as a PE stabilizer in the food and pharmaceutical industries and related fields.

## Figures and Tables

**Figure 1 pharmaceutics-14-01561-f001:**
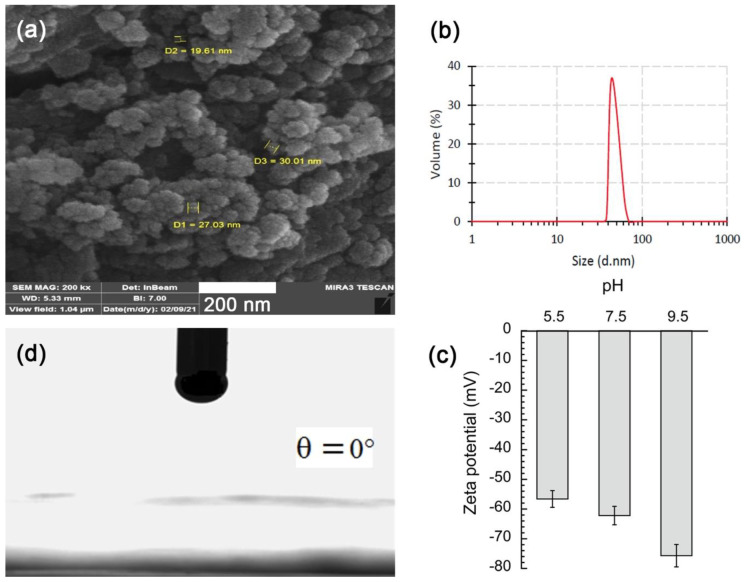
Characterization of native SiO_2_ NPs. (**a**) SEM picture with (**b**) the corresponding size distribution obtained from DLS, and (**c**) measurement of zeta potentials at different pHs. (**d**) Contact angle of native SiO_2_ NPs.

**Figure 2 pharmaceutics-14-01561-f002:**
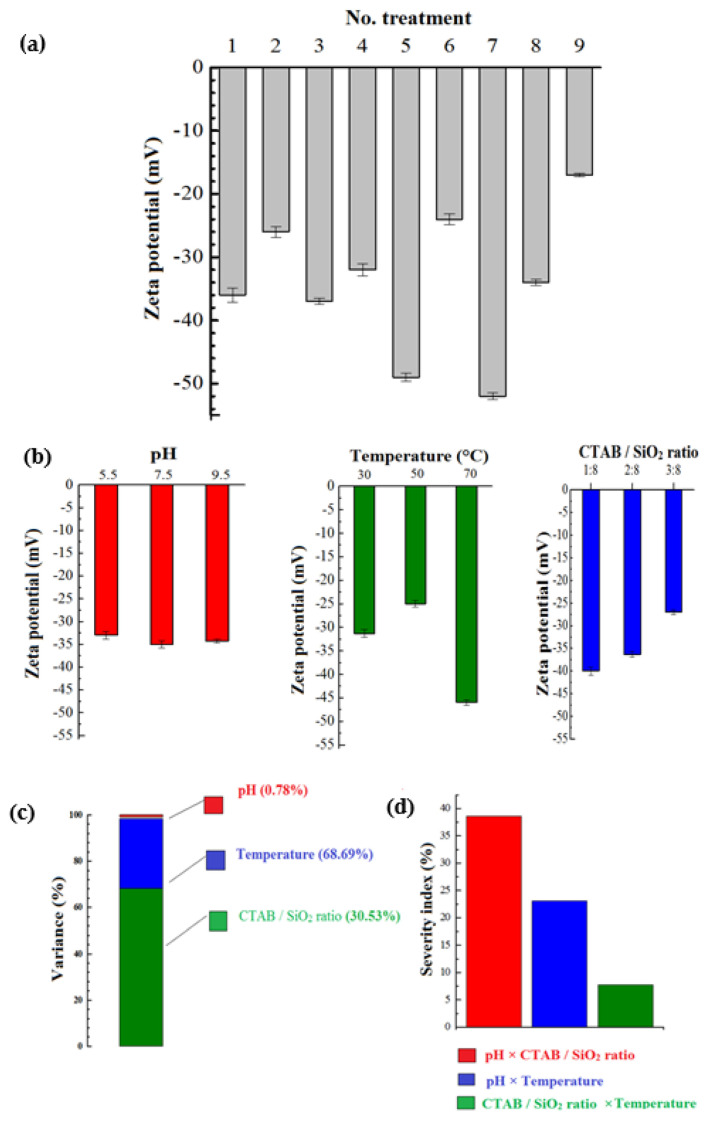
Characterization of the surface modification of SiO_2_ nanoparticles. (**a**) zeta potential of modified NPs according to different treatments detailed in Table 1. (**b**) Mean values of zeta potentials of individual factors, completed with the values of (**c**) variance and (**d**) severity index.

**Figure 3 pharmaceutics-14-01561-f003:**
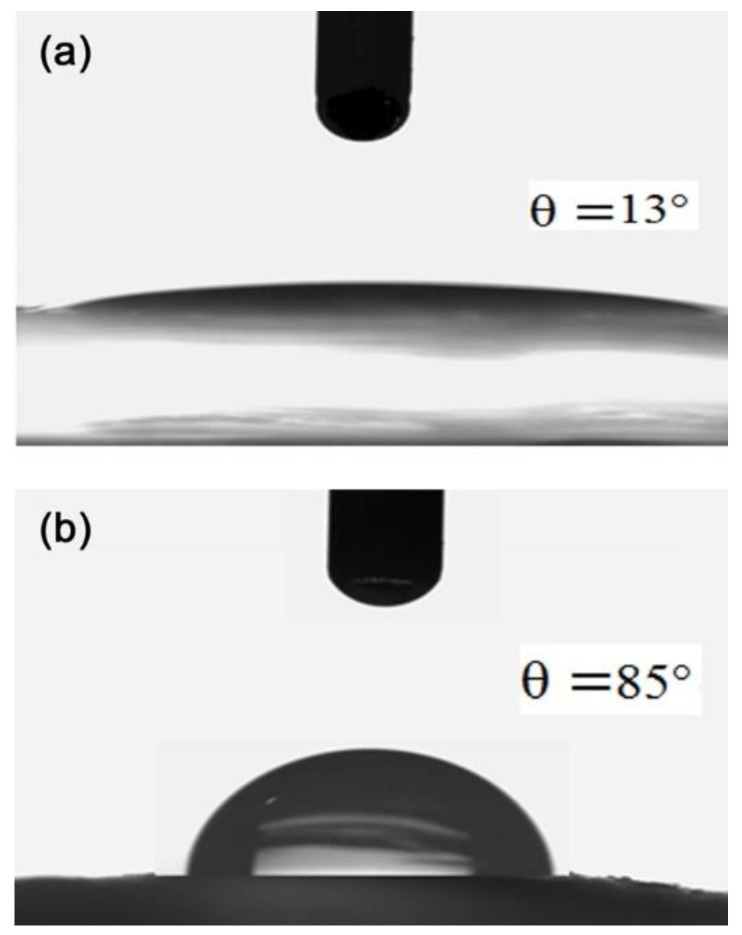
Contact angle *θ* for water on a surface coated with SiO_2_ NPs after (**a**) treatment No. 7, and (**b**) treatment No. 2.

**Figure 4 pharmaceutics-14-01561-f004:**
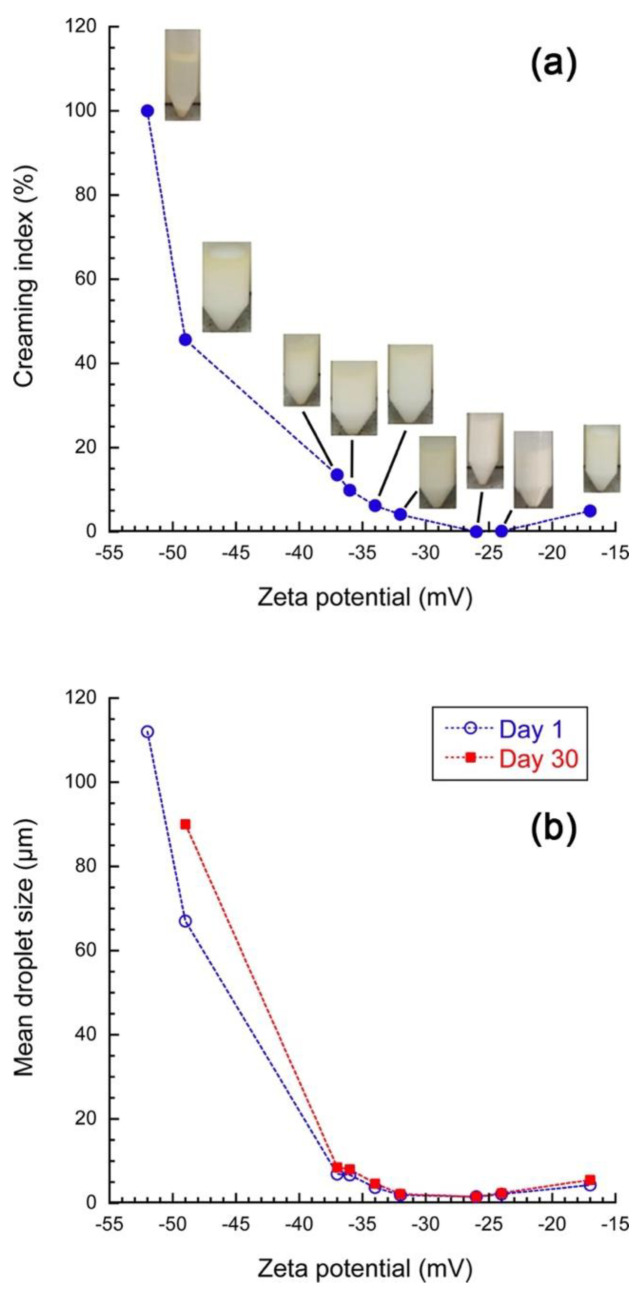
Characterization of PEs stabilized by modified SiO_2_ NPs. (**a**) Creaming index with corresponding images at 30 days and (**b**) coalescence of PEs, both as a function of the zeta potential of MSNs.

**Figure 5 pharmaceutics-14-01561-f005:**
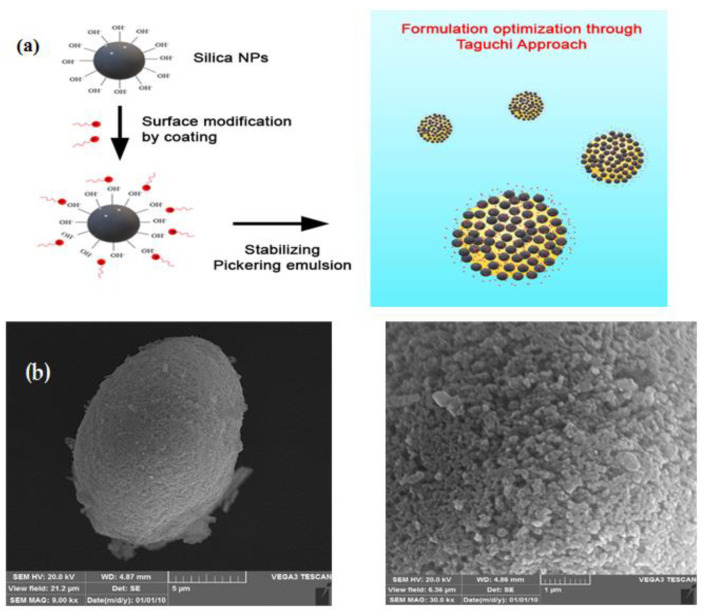
SiO_2_ NPs’ surface modification process and formation of Pickering emulsion stabilized by MSNs (**a**). SEM characterization of Pickering emulsions stabilized with SiO_2_ NPs (**b**).

**Figure 6 pharmaceutics-14-01561-f006:**
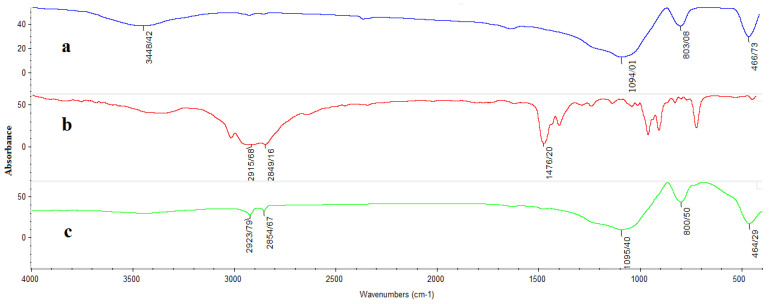
FTIR spectra of (**a**) SiO_2_ NPs, (**b**) CTAB, and (**c**) MSNs.

**Figure 7 pharmaceutics-14-01561-f007:**
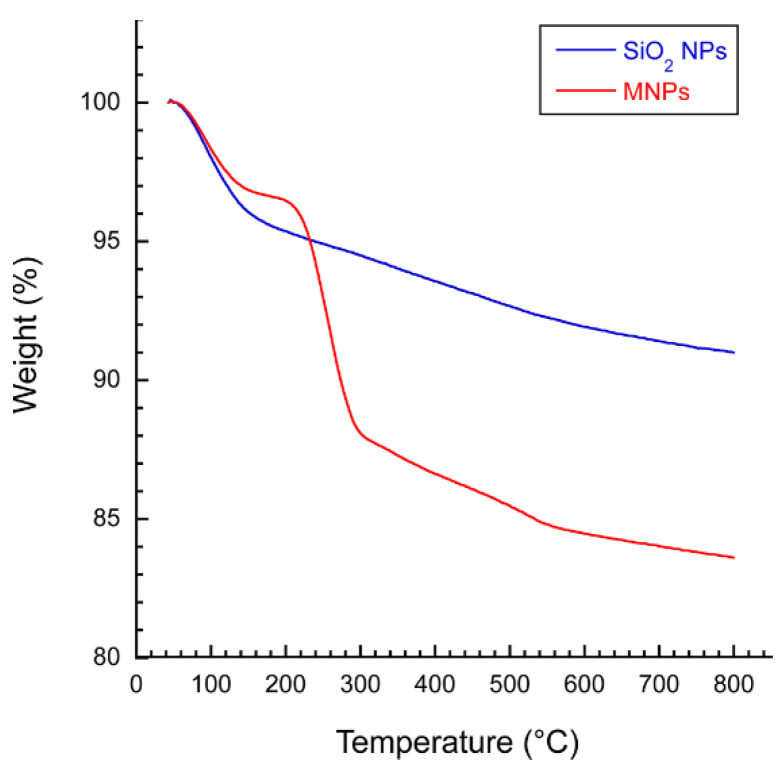
TGA thermograms of native (SiO_2_ NPs) and modified NPs (MSNs).

**Table 1 pharmaceutics-14-01561-t001:** Experimental design according to the Taguchi method.

Treatment No	pH	Surfactant/SiO_2_ NPs Ratio	Temperature (°C)
1	5.5	1:8	30
2	5.5	2:8	50
3	5.5	3:8	70
4	7.5	1:8	50
5	7.5	2:8	70
6	7.5	3:8	30
7	9.5	1:8	70
8	9.5	2:8	30
9	9.5	3:8	50

## Data Availability

The data that support the findings of this study are available from the corresponding author, upon reasonable request.

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
