# Peer review of "Preparation of Pickering Emulsions Stabilized by Modified Silica Nanoparticles via the Taguchi Approach"

_pharmaceutics, 2022, doi:10.3390/pharmaceutics14081561_

Round 1
Reviewer 1 Report
In figure 1b the caption needs to be updated (if it was really the DLS curve which is shown!)
Author Response
“In figure 1b the caption needs to be updated (if it was really the DLS curve which is shown!)”
Answer: Thank you for the comment, it is effectively DLS curve, this mistake is now corrected in the revised manuscript.
Reviewer 2 Report
Attached please see the suggestions, thank you!

Author Response
“(1) The authors should re-organize the presentation, to deliver the novelty they claimed. Currently, it’s extremely difficult for me to get it.”
Answer: This point was clarified in the revised manuscript. Indeed, the originality of this study lies in the experimental approach carried out, to use and optimize SiO2 nanoparticles, as an efficient solution to stabilize Pickering emulsions. The multistep process, from surface modification of SiO2 NPs, the formulation of Pickering NEs, up to the optimization of their stability according to the Taguchi method, also constitutes a new insight into that field of pharmaceutical formulation. The sentence was added in the introduction to clearly present the message to the reader, p.2, lines 81 to 86.
----------
“(2) All the figures should be re-made. You can refer to ACS journals how to prepare high-quality figures.”
Answer: Figure 1 was reorganized and redrawn with higher definition, all other figures are carefully checked to correspond to the usual standards for scientific publications, in term of resolution and presentation.
Round 2
Reviewer 2 Report
It's ok for publish now.
This manuscript is a resubmission of an earlier submission. The following is a list of the peer review reports and author responses from that submission.
Round 1
Reviewer 1 Report
Dear Authors,
The manuscript presents an experimental research on the oil-in-water Pickering emulsions stabilized by modified silica nanoparticles. Authors carried out several measurements such as: zeta potential, FTIR, TGA, wettability, morphological as well as droplet size tests, physical stability study and statistical analysis. The methods used in the research are clear and understandable. Undoubtedly the authors master their approach well and obtained a lot of interesting results. However, below are questions and remarks that should be corrected before the manuscript could be consider for publication in Pharmaceutics:
This paper can be published after MINOR REVISIONS.
- How many scans were performed in each FTIR measurements? (Paragraph 2.6)
- How many measurements of each sample were performed in TGA study? (Paragraph 2.9)
- Please add the significance level in Paragraph 2.12.
- The notes in the picture in Fig. 1a are invisible.
- Please add the error bars in Fig. 1 b and c as well as in Fig.2.
- In Fig.2 mark statistically significant differences.
- Axes labels in Fig. 1b, Fig. 2b, c, d are illegible.
- The description below Fig. 2d is difficult to read.
- The FTIR spectra in Fig. 6a is cut off at the top at about 800-500 cm-1.

Reviewer 2 Report
In this manuscript, the authors, Fatemeh Heidari et al investigated the stabilization of PEs by SiO2 NPs, as a function of the surface modification of silica NPs. For this modification the Taguchi approach was followed, by coating the SiO2 NPs surfaces with the CTAB surfactant. These were then tested under various temperatures and at different pH values. The authors concluded that, in general, the optimal surface modification conditions of the SiO2 NPs reported in this work might be useful as PE stabilizers in the food and pharmaceutical industries and other related fields, as well.
The manuscript is well written and the results nicely presented.
Reviewer 3 Report
Please see the attachment, thank you!

Reviewer 4 Report
The behavior of particles and surfactants in pickering emulsions is attracting attention from various scientific fields. In this study, the behavior of Pickering emulsions composed of cationic surfactant (CTAB) and silica nanoparticles is approached from a physically-chemical viewpoint. Obtained results provide valuable and useful information. Reviewer felt that the results obtained in this report are valuable for researchers in this area, but the reviewer suggest the following comments.
Comment 1
In this study, CTAB is used. Explanation should be added in the text as to why this surfactant was selected among the various surfactants.
Comment 2
Reviewer recommend the addition of one new figure of illustration (Pickering emulsion composed of CTAB and silica nanoparticles) that describes the overall concept of this study. This new figure would help the reader better visualize the phenomena revealed in this paper.
Comment 3
Authors mentioned that CTAB and SiO2 nanoparticles are biocompatible and are used in pharmaceuticals. On the other hand, several reports have raised concerns about their safety. Thus, the text seems to be lacking in explaining the safety of CTAB and SiO2 nanoparticles to the reader. The following review supports the biocompatibility of the materials that the authors used in the experiment. Such papers should be cited to emphasize the low toxicity of the materials comprising the pickering emulsions in this study.
Int J Nanomedicine . 2021;16:3937-3999.
A Critical Review of the Use of Surfactant-Coated Nanoparticles in Nanomedicine and Food Nanotechnology
https://pubmed.ncbi.nlm.nih.gov/34140768/
Minor comment
Line 35: “SiO2 nanoparticles” should be subscript. “SiO2 nanoparticles”.
Reviewer 5 Report
This manuscript is about fabricating particle-stabilized sunflower-oil-in-water emulsions using silica particles somewhat hydrophobized using CTAB - a topic I am very well familiar with, since we have made in the past pretty much comparable emulsions using silicone oil and Stöber silica, hydrophobized with CTAC. Therefore I was really looking forward to reading the manuscript and I had a quite positive attitude, because I like the topic.
I would say in general this could have been a nice piece of work, but in my opinion there are a few serious flaws which need to be corrected before this manuscript can be published – therefore I recommend at least „major revision“, but including additional experiments!
What are my "objections"?
Lines 152ff: Size distribution of your silica NPs: How could you possibly tell from SEM images that the particles you observe are INDIVIDUAL nanoparticles of about 30 nm? Are you sure that by the sintering process, the nanoparticles are not partially „merging“, forming considerably larger „objects“ consisting of primary particles connected by „sinter necks“. One famous examples of such „coral-like“, not at all spherical particles is fumed silica. Why is this important? Because a spherical Stoeber silica particle can be expected to „find“ the oil/water interface without any problem, whereas fumed silica („fluffy“ 3D-object of several hundred nm in size) does not … Why didn’t you measure the particle size distribution of your silica NPs by using DLS (Malvern Nanosizer)? You seem to have this equipment and it should be ideally suitable to answer this question! Therefore: Please replace Figure 1b by a DLS measurement!
Lines 173ff: You did not say anything HOW you modified your silica NP with CTAB: Did you add concentrated CTAB, or a predilution? Adding a too concentrated CTAB solution to silica can lead to an inhomogeneous population: silica NP adsorbing a lot of CTAB and turning pretty hydrophobic, next to the rest of the NPs remaining rather hydrophilic …
Lines 191ff: I do not understand your argument „higher amount of CTAB molecules is converted to micelles in water due to dependence of critical micelle concentration with temperature.“ The solubility of ionic surfactants should be better at elevated temperatures, which is just die opposite effect: Faster Brownian motion, higher solubility of CTAB leading to better surface modification … Maybe the surface modification was just „more homogeneous“ (see my comment to lines 173ff) at 70°C?
Line 187: 68,688%: Wow! How precise are your ZP measurements?? ?
Lines 230-235: This paragraph is unrelated to modifying silica and does not help to understand the system studied – that’s why I would recommend to delete this portion (incl. Ref. 43)
Line 250 (Figure 4): You claimed that you have measured the droplet sizes of your Pickering emulsions by using DLS – I found this statement already in line 124 quite surprising, because according to my experience Pickering emulsions are often rather coarse. So how did you measure the particles sizes really? DLS is not an option, because some of the data points (110, 90, 70 µm ) are way outside of the accessable range of DLS, which goes under ideal conditions up to 10 µm …
Line 301 (Figure 5): This is really an image of a sunflower oil droplet in vacuum of the SEM? Sorry, I am not an electron microscopy expert, but I find this somewhat surprising …
Reference 29 in incomplete and should be: Suttiruengwong, S.; Puathawee, P.; Chareonpanich, M. Preparation of mesoporous silica from rice husk ash: Effect of depolymerizing agents on physico-chemical properties. Adv. Mater. Res. 2010, 93–94, 664–667.